# Grain Quality Characterization of Hybrid Rice Restorer Lines with Resilience to Suboptimal Temperatures during Filling Stage

**DOI:** 10.3390/foods11213513

**Published:** 2022-11-04

**Authors:** Xuedan Lu, Lu Wang, Yunhua Xiao, Feng Wang, Guilian Zhang, Wenbang Tang, Huabing Deng

**Affiliations:** 1Hunan Key Laboratory of Disease Resistance Breeding of Rice and Rape, College of Agronomy, Hunan Agricultural University, Changsha 410128, China; 2Hunan Hybrid Rice Centre, Hunan Academy of Agricultural Science, Changsha 410125, China

**Keywords:** rice restorer lines, grain quality, starch, temperature stress tolerance

## Abstract

Rice (*Oryza sativa* L.) is a staple food that is consumed worldwide, and hybrid rice has been widely employed in many countries to greatly increase yield. However, the frequency of extreme temperature events is increasing, presenting a serious challenge to rice grain quality. Improving hybrid rice grain quality has become crucial for ensuring consumer acceptance. This study compared the differences in milling quality, appearance quality, and physical and chemical starch properties of rice grains of five restorer lines (the male parent of hybrid rice) when they encountered naturally unfavorable temperatures during the filling period under field conditions. High temperatures (HTs) and low temperatures (LTs) had opposite effects on grain quality, and the effect was correlated with rice variety. Notably, R751, R313, and Yuewangsimiao (YWSM) were shown to be superior restorer lines with good resistance to both HT and LT according to traits such as head rice rate, chalkiness degree, chalky rice rate, amylose content, alkali spreading value, and pasting properties. However, Huazhan and 8XR274 were susceptible to sub-optimal temperatures at the grain-filling stage. Breeding hybrid rice with adverse-temperature-tolerant restorer lines can not only ensure high yield via heterosis but also produce superior grain quality. This could ensure the quantity and taste of rice as a staple food in the future, when extreme temperatures will occur increasingly frequently.

## 1. Introduction

Rice (*Oryza sativa* L.) is a staple food for more than half of the world’s population [1]. Improving rice grain quality has become crucial for ensuring consumer acceptance due to the growing demand for high-quality rice [2]. However, grain quality is influenced significantly by the field environment, and in particular, by air temperature [3,4,5,6,7,8,9,10]. Extreme temperature events (including high and low temperatures) will likely increase in frequency and intensity under the conditions of global warming, presenting a serious challenge for future rice yield and grain quality [11]. Hence, cultivation of high-quality rice with superior temperature resilience is a priority for breeders.

Rice grain quality covers the physicochemical properties of rice kernels that influence the milling, appearance, cooking, taste, and nutritional properties [2]. Milling quality, along with yield, determines the edible yield and economic value [12]. The appearance of rice grains is primarily evaluated based on transparency. Chalkiness is one of major issues with appearance. More importantly, it negatively affects milling quality and results in poorer eating and cooking quality (ECQ) [13]. The grain quality relies on the microstructure and physicochemical properties of the starch that accounts for over 80% of the grain weight. Starch is a mixture of two homopolymers, amylose (linear α-1, 4-polyglucans) and amylopectin (α-1, 6-branched polyglucans) [14,15]. The amylose content (AC), chain length distribution (CLD) of amylose and amylopectin, gel consistency (GC), gelatinization temperature (GT), alkali spreading value (ASV), and pasting viscosity are widely used to assess the physiochemical properties of starch and indirectly evaluate the ECQ of rice [2,7,15,16,17,18,19].

Air temperature during the growing season, especially during grain filling, plays important roles in endosperm development and grain quality. The typical symptoms resulting from high temperature (HT) at the filling stage include increased grain-filling rate, poor milling quality, increased chalkiness, low amylose content, an altered amylopectin chain length distribution, and other changes in starch properties, such as loosely packed and irregular starch granules [5,6,7,9,20,21,22,23,24,25,26]. Low temperature (LT) treatment prolongs the grain-filling period, increases the amylose content, and affects the activities involved in starch biosynthesis [27]. In addition, a LT during the grain-filling stage results in the deterioration of cooked rice quality, and this poor palatability of cooked rice could be attributed to high amylose content, low pasting viscosity, and reduced GC [8,28]. These studies suggest that the adjustment of sowing date can reduce the negative influence of temperature stress on rice quality.

Although many studies on the effects of high or low temperatures during grain-filling on rice quality have been conducted, most of this work has only focused on one form of temperature stress [5,6,7,8,9,20,21,22,23,24,25,26,28]. In the context of frequent extreme weather, a rice variety may experience both HT stress and LT stress during grain filling in different years. However, there is a lack of research on the identification of high-quality rice varieties resistant to both HTs and LTs. Hybrid rice takes advantage of heterosis (hybrid vigor), a phenomenon where F_1_ hybrids derived from crosses between genetically distinct inbred varieties exhibit superior phenotypic performance compared to their parents [29]. As rice is a self-pollinated species, hybrid rice breeding uses sterile male lines as female parents and distantly related restorer lines as male parents. Such hybrids have played an increasingly important role in ensuring food security with high yield [30]. The restorer line provides a sperm nucleus that combines with the two polar nuclei provided by the male sterile line to form an endosperm nucleus, which then develops into endosperm, the predominant object of study in rice quality. Thus, it is reasonable that restorer lines with stable and advantageous quality characteristics will provide genetic information for endosperm development and contribute to the grain quality in hybrid progeny. We believe that compared with adjusting the sowing date to escape temperature stress, the identification of restorer line varieties with tolerance to both high and low temperatures would better ensure the quality of rice grains and the hybrid rice varieties derived from such lines. However, there is little information concerning restorer varieties with good rice quality and insensitivity to adverse temperatures at the grain-filling stage.

Here, differential responses of five restorer lines to both HT and LT at the grain-filling stage were investigated by conducting field experiments with HT, LT, and normal temperature (NT) treatments implemented by setting the sowing date. Field experiments can better embody climate variation than those conducted in artificially controlled indoor conditions for actual rice production. Rice milling quality, appearance quality, starch composition, granule characters, and pasting viscosity were measured to identify restorer lines whose rice grain quality was least impaired by either high or low temperature stress. Changes in physical and chemical properties of rice grains caused by temperature stress are likely to lead to deterioration of the taste of various rice foods, such as cooked rice, rice noodles, steamed rice sponge cake, and rice crust. The aim of this work was to identify the restorer lines that have excellent grain quality and that are insensitive to both high temperatures and low temperatures. The results will contribute to creating germplasm resources to ensure the taste of rice is unaffected by temperature stress.

## 2. Materials and Methods

### 2.1. Plant Materials and Experimental Design

Five *indica* varieties that could be utilized as restorer lines for hybrid rice were used as plant materials for this experiment. Yuewangsimiao (YWSM) and Huazhan have been widely popularized in hybrid rice breeding in China. R751, R313, and 8XR274 are three newly cultivated restorer lines derived by our research group, and the hybrid rice variety Tangliangyou751 created by R751 has been approved by the Chinese government and has the potential to be grown in a wide area.

The field experiments were conducted in 2020 at the experimental field of Hunan Agricultural University, in the subtropical environment of Changsha City, Hunan Province, China (28°12′ N, 112°59′ E, 41 m altitude). The rice materials were sown in six batches, with different sowing dates: 11 April, 25 April, 9 May, 23 May, 6 June, and 20 June. The climate data regarding daily radiation and air temperature were measured using a Field Temperature and Humidity Recorder (L92-1, Luge, Hangzhou, China). The dates of sowing, panicle initiation, heading, and maturity were recorded for determining growth duration. According to the meteorological data and the corresponding developmental stage of the variety, three batches were chosen for each variety with three grain-filling temperatures, namely, LT (implemented by setting a late sowing date, average temperature ≈ 21.2 °C), NT as a control (implemented by setting an early sowing date, average temperature ≈ 26.2 °C), and HT (implemented by setting an earlier sowing date, average temperature ≈ 30.3 °C). As shown in Table 1, each variety was treated with LT, NT, and HT at the grain-filling stage.

The experiments were conducted in a randomized block pattern with three replications. Twenty plants were planted in each plot, with a row spacing of 20 cm × 20 cm. Weeds, pests, and diseased plants were intensively controlled. After ripening, 2 kg of rice grains were collected and dried to about 13% moisture. The rice was stored at room temperature for three months for the determination of rice quality and RVA characteristics of the starch.

### 2.2. Determination of Milling Quality

Rice kernel samples (20–25 g) (m_0_) were weighed (accurate to 0.01 g), and the germinated grains were picked out. The germinated grains were dehusked separately and weighed to record the mass of brown rice of germinated grains (m_1_). The remaining samples were shelled by a huller, and the mass of brown rice was measured and recorded (m_2_). The incomplete brown grains were picked out by sensory inspection, and the mass of incomplete brown grains was measured (m_3_). The brown rice rate was calculated according to Formula (1):(1)$$\;\mathrm{Brown}\;\mathrm{Rice}\;\mathrm{Rate}=\frac{\left(m_{1}+m_{2}\right)-\left(m_{1}+m_{3}\right)/2}{m_{0}}\times 100$$

The mass of samples obtained in the previous step was measured (m_3_), and the samples were poured into the milling chamber, and the chamber was adjusted to the optimal milling time to make the accuracy of milling reach the third level of the national standard. The milled rice grains were sieved through a diameter of 1.5 mm to remove the embryo and bran. After cooling to room temperature, the mass of milled rice (m_4_) was measured (accurate to 0.01 g). The milled rice rate was calculated according to Formula (2):(2)$$\;\mathrm{Milled}\;\mathrm{Rice}\;\mathrm{Rate}=\frac{m_{4}}{m_{3}}\times \mathrm{Brown}\;\mathrm{Rice}\;\mathrm{Rate}$$

An appearance detection analyzer (MRS-9600TFU2L, MICROTEK, Shanghai, China) was used to collect the sample images. The image analysis system automatically analyzed and judged the image information in the sample. The mass fraction of head rice in milled rice grains (w) was then calculated. The head rice rate was calculated according to Formula (3):(3) Head Rice Rate=w×Milled Rice Rate/100

### 2.3. Chalkiness Degree and Chalky Rice Rate Determination

One hundred grains were chosen randomly from the milled rice sample and placed on a chalkiness observation instrument (SC-E, Wseen, Hangzhou, China). Grains with chalkiness (termed as grains with a white belly, white center, and white back or a combination of these) were counted. The chalky grain rate was calculated as the percentage of grains with chalkiness. Ten intact milled rice grains were chosen randomly from the grains with chalkiness. The average percentage of chalky area in the plane projection of the whole rice grain was measured using visual observation. The chalkiness degree was calculated according to Formula (4):(4)$$\mathrm{Chalkiness}\;\mathrm{degree}\left(\%\right)=\;\mathrm{chalky}\;\mathrm{grain}\;\mathrm{rate}\;\left(\%\right)\times \mathrm{average}\;\mathrm{chalky}\;\mathrm{area}\;\left(\%\right)/100.$$

### 2.4. Starch Isolation

Milled rice samples were stored in sealed bags under refrigeration (4 °C) until analysis. The polished rice was ground into flour in a mill (FOSS 1093 Cyclotec Sample Mill, Hoganas, Sweden) with a 0.5 mm screen, and starch was isolated as described previously [9].

### 2.5. Scanning Electron Microscopy (SEM)

Brown rice grains were cracked with a razor blade, and the transverse section surface was coated with gold for 90 s using a vaporizer and observed under a scanning electron microscope (SU8010, Hitachi, Tokyo, Japan). Observation conditions were as follows: acceleration voltage, 2000 V; magnification, 10,000.

### 2.6. Starch Size Distribution and Chain Length Distribution (CLD) Analysis

For starch size distribution analysis, the samples were removed after water balance, ground, and dispersed with a mortar. Then, they were passed through a 200-mesh sample sieve. A sample of 100–200 mg of starch was placed in a clean Eppendorf tube, and then 1 mL 75% alcohol was added to the Eppendorf tube and mixed well. The samples were measured with dynamic light scattering (Mastersizer 3000, Malvern Panalytical, Malvern, UK). All tests were performed in triplicate.

For CLD analysis, a sample of 5 mg purified starch suspended in 5 mL water was subjected to boiling water bath for 1 h. Sodium acetate (50 μL, 0.6M, pH 4.4), sodium azide (10 μL, 2% *w*/*v*), and isoamylase (10 μL, 1400U) were added and incubated at 37 °C for 24 h before mixing with sodium borohydride solution (0.5% (*w*/*v*)). After another 20 h incubation, a sample of 600 μL was blown dry, then dissolved in sodium hydroxide (30 μL, 1M) for 1 h, and diluted with water (570 μL). The supernatant after centrifugation can be loaded on a chromatographic system (ICS500+, Thermo Fisher Scientific, Waltham, MA, USA) with a liquid chromatographic column (Dionex ™ CarboPac ™ PA200) and the electrochemical detector to quantitatively analyze the CLD based on the Glucan Standards (Oligosaccharides Kit, S47265, Sigma, Buenos Aires, Argentina).

### 2.7. Determination of the Amylose Content (AC), Gel Consistency (GC), and the Alkali Spreading Value (ASV)

AC of the flour was determined following a modification of the iodine colorimetric method described by Man et al. [31]. After water balance, 0.1000 ± 0.0002 g of rice flour sample and the standard samples were fully wet by ethanol (1.0 mL) and mixed with sodium hydroxide solution (1.0 M, 9 mL) before incubation overnight (37 °C). By adding iodine potassium iodide solution (0.2%) for color development, the OD_620_ value was measured. Based on the standard curve performed according to AC in the standard samples, AC in the tested sample was calculated.

GC was measured according to a method described by Tan et al. with minor revision [32]. The crushed milled-rice was passed through a 100-mesh sieve and then 87.8–88.2 mg dry sample was carefully put into a round bottom test tube (inner diameter: 11 mm; length: 100 mm). After adding phenol blue indicator (0.2 mL) and potassium hydroxide solution (0.20 M, 2.0 mL), the test tube was covered with a glass ball and subjected to a boiling water bath for 8 min, room temperature for 5 min, an ice water bath for 20 min, and then incubation flatly for 1 h (temperature: 25 ± 2 °C; humidity: 60 ± 5%). The length measured from the bottom of the tube to the front edge of the rice glue was the GC value.

ASV was measured visually by soaking the milled rice grains in 1.4% KOH solution for 24 h at 30 °C as described by Mariotti et al. [33]. The milled rice grains should be without breakage or cracks, and with uniform size and maturity. The endosperm appearance and digestion diffusion degree were visually inspected. The grading of each rice grain was evaluated one by one mainly based on the decomposition degree.

### 2.8. Pasting Property Measurement

Rice flour (3.0 g) stirred in 25 mL water was used to measure RVA characteristics, including pasting temperature (PT), peak viscosity (PKV), hot paste viscosity (HPV), cool paste viscosity (CPV), setback viscosity (SBV), and breakdown viscosity (BDV). The analyses used a Rapid Visco Analyzer (Newport Scientific Pty, Warriewood, NSW, Australia) [17].

### 2.9. Water Solubility Index and Swelling Power Assay

Swelling power and water solubility were determined according to a previous method [9]. Starch samples (m_0_) were mixed with water (2%, *w*/*v*), placed in a 2 mL centrifuge tube (m_1_), and heated in a water bath at 95 °C for 30 min. The tubes were gently shaken for 1 min. The samples were cooled to room temperature, centrifuged at 8000× *g* for 10 min, and the supernatant was discarded. The colloid remaining in the centrifuge tube was weighed (m_2_), and the sediments were dried to constant weight (m_3_) at 60 °C. The swelling power and solubility were calculated as follows: swelling power = (m_2_ − m_1_)/(m_3_ − m_1_) (g/g); solubility (%) = 100 (m_0_ + m_1_ − m_3_)/m_0_ × 100%.

### 2.10. Statistical Analysis

All parameters shown in the tables and figures used in this article represent the mean values of the experimental data obtained from triplicate tests for all varieties sown during the three planting periods. The analysis of all data was performed using the SPSS16.0 Statistical Software. Two-way analysis of variance and Tukey’s tests were used to determine whether statistically significant differences (*p* < 0.05) existed between the means.

## 3. Results and Discussion

### 3.1. Milling Quality

Milling quality is usually evaluated by indicators such as brown rice rate, milled rice rate, and head rice rate. To examine the effect of improper temperatures at the grain-filling stage on the milling quality of grains from restorer lines, we measured and compared the three indicators of the grains harvested after different temperature treatments. As shown in Figure 1A, the brown rice rate of the restorer line R751 was affected neither by HT nor LT; and YWSM, 8XR274, and Huazhan exhibited unchanged brown rice rates upon HT treatment at the grain-filling stage. However, LT decreased the brown rice rate of the latter three restorer lines by 2.30%, 2.04%, and 1.77%, respectively (Figure 1A). R313 was shown to be the most vulnerable variety among those tested, as its brown rice rate was reduced significantly by LT (3.63%) and increased by HT (2.69%). It was shown that LT during the grain-filling stage had a greater negative effect than HT on the brown rice rate of most varieties except for the resistant line R751.

HT increased the milled rice rates of R751, R313, YWSM, and Huazhan significantly by 2.02%, 3.11%, 1.15%, and 0.98%, respectively, but it did not change that of 8XR274 (Figure 1B). The milled rice rate of all five varieties was reduced by more than 2.15% by LT (Figure 1B). The results suggested that LT during the grain-filling stage led to a decline, and HT resulted in an increase in the milled rice rate; however, LT had a greater impact for most tested varieties.

Head rice is conventionally defined as intact grains that have 3/4 of the original kernel length after complete milling [13]. Head rice yield (HRY) is the gold standard of rice millers to quantify milling quality [13]. High temperature decreased the head rice rates of 8XR274 and Huazhan by 31.3% and 26.67%, respectively (Figure 1C). Although the high temperature also reduced the head rice rates of YWSM (by 6.63%) and R313 (by 1.87%), their degrees of reduction were much lower than those of 8XR274 and Huazhan (Figure 1C). Likewise, compared with the significant negative impact of LT on 8XR274 (by 38.77%) and Huazhan (by 11.39%), head rice rates of R751, R313, and YWSM exhibited only slight decreases (by 3.14%, 3.24%, and 4.95%, respectively) with LT (Figure 1C). Interestingly, differently from the opposite effects of high and low temperatures on milled rice rate, both HT and LT reduced the head rice rate (Figure 1C). Our results demonstrated that considering HRY, the varieties R751, R313, and YWSM are more resistant to HT and LT, whereas 8XR274 and Huazhan were more susceptible to improper temperatures. The results suggested that improper temperature-caused loss of HRY for R751, R313, YWSM, could be less than that of 8XR274 and Huazhan.

### 3.2. Appearance Quality

Chalk, as the inverse indicator of the appearance quality, is usually induced by elevated temperature during grain filling [20,25,26,34], and it negatively affects milling quality and ECQ [2]. However, whether LT affects chalkiness formation is not particularly clear. To investigate the effects of HT and LT on chalk and identify superior restorer lines with stable good appearance, we examined the appearance of grains from five restorer lines experiencing low and high temperatures at grain-filling stages, with quantitative measurements of chalky grain rate and chalkiness degree. As shown in Figure 2A, the grains of R751, R313, and YWSM were more translucent than those of 8XR274 and Huazhan under normal temperature (NT). Indeed, the chalkiness degrees of R751, R313, and YWSM were about 1.1–2.1%, and the chalkiness degrees of 8XR274 and Huazhan were as high as 23.7% and 9.9%, respectively (Figure 2C). The chalky grain rates of R751, R313, and YWSM were only 5.6–7.7%, whereas those of 8XR274 and Huazhan were as high as 83.6% and 21.4%, respectively (Figure 2B). R751, R313, and YWSM demonstrated better appearance quality than 8XR274 and Huazhan under NT.

Furthermore, under the LT at the grain-filling stage, the chalkiness degree and chalky grain rate of 8XR274 were decreased by 29.55% and 21.52%, respectively (Figure 2B,C). LT decreased the chalkiness degree and chalky grain rate of Huazhan by 32.24% and 65.66%, respectively (Figure 2B,C). Our results demonstrated that the LT caused a reduction of chalkiness, indicating the positive effect of LT on the appearance quality of sensitive varieties. Previous studies reported that unusual starch degradation, rather than starch synthesis, was involved in the occurrence of chalky grains of rice [35]. LT might debase the activity of enzymes involved in starch degradation in the varieties sensitive to LT, thereby weakening the starch degradation process and leading to the reduction of chalkiness.

Moreover, under HT at the grain-filling stage, the chalkiness degree and chalky grain rate of 8XR274 were increased by 84.39% and 14.83%, respectively (Figure 2B,C). HT increased the chalkiness degree and chalky grain rate of Huazhan dramatically by 106.06% and 136.45%, respectively (Figure 2B,C). The results showed that HT further caused grain chalkiness increases in varieties with poor appearance quality, which is consistent with previous research on conventional rice or hybrid rice combination varieties [9,20,21,22,23]. However, our research further identified three restorer lines of hybrid rice whose appearance quality was unaffected by HT or LT. The chalkiness degree or chalky grain rates of R751, R313, and YWSM under HT and LT showed no significant differences compared with those under NT (Figure 2). Overall, R751, R313, and YWSM possessed better appearance quality than the other two restorer lines, and these desirable characteristics were unlikely to be affected by temperature stress.

### 3.3. Starch Granule Morphology and Granule Size Distribution

The formation of chalk usually accompanies abnormal starch granule morphology, including incompletely filled starch granules with many air spaces in between that are visible as opaque spots along the translucent grain [36]. In order to intuitively observe the morphological characteristics, including the arrangement compactness of starch granules in rice grains, we examined horizontal slices of grains via scanning electron microscopy instead of isolated starch obtained after the rice grains were ground into powder, as reported in a previous study [4]. In the grains of R751, R313, and YWSM treated with LT, NT, or HT at the filling stage, the starch granules were semi-crystalline and tightly packed, and the shapes of starch granules were polygonal with sharp edges (Figure 3). However, in the grains of 8XR274 and Huazhan, the starch granules were loosely packed with many air spaces and were nearly round in shape, lacking edges and corners, for each temperature treatment that the grains experienced during endosperm development (Figure 3). More air spaces in between starch gains and many small spherical granules existed in 8XR274 grains under HT. Dense pits were frequently observed on the surfaces of starch granules in Huazhan grains grown under HT (Figure 3), which is consistent with previous studies concerning the effect of high temperatures on starch granules [26,35]. However, the influence of LT on starch granule morphology of 8XR274 and Huazhan was not as strong as that of HT (Figure 3). This is consistent with our previously observed effects of LT on the chalkiness degrees of 8XR274 and Huazhan, being less than those of HT to some extent (Figure 2). HT and LT increased the chalkiness degrees of 8XR274 by 84.39% and 27.42%, respectively. For Huazhan, HT and LT increased the chalkiness degree by 106.06% and 65.66%, respectively (Figure 2). As a qualitative technique, using SEM analysis it was difficult to show the effect of LT on the morphology of starch particles (Figure 3). Overall, the fine starch granule morphologies of R751, R313, and YWSM were unaffected by suboptimal temperatures, whereas the starch granules of 8XR274 and Huazhan exhibited abnormal morphology, especially under HT.

To further characterize the effect of temperature on the size distribution of starch panicles from various restorer lines, we compared the differences in starch volume distributions (Appendix A). Upon HT treatment, the curve peak value of the 8XR274 particle volume distribution increased by 39.19% with a narrower curve range, whereas LT did not show a significant effect on the parameter (Appendix A). Moreover, the curve peak shifted to the left slightly (6.61–8.78 μm to 5.75–7.59 μm) under HT, suggesting that HT led to the formation of more small starch granules in 8XR274. The results were likely to be consistent with the SEM analysis (Figure 3). It was demonstrated that HT rather than LT exerted an effect on the starch size distribution of 8XR274.

Nevertheless, the curves of R751, YWSM, and Huazhan under HT nearly coincided with their counterparts under NT, indicating that the starch particle sizes of these three restorer lines may be unaffected by HT (Appendix A). In R313, although the peak value under HT was 11.37% higher than that under NT, the curve was not shifted, indicating that HT had limited influence on the starch size of R313. However, for LT, we found that the curve ranges were wider and peak values were decreased by 31.61%, 21.43%, and 26.79% for R751, R313, and YWSM upon LT treatments, respectively (Appendix A). The results suggested that LT, rather than HT, affected the starch granule size distribution of R751, R313, and YWSM. Notably, Huazhan was the only variety with a starch particle size distribution unaffected by either HT or LT. However, the peak value from Huazhan was clearly lower than those of the other four varieties, and this may have contributed to the widely different morphological features (Figure 3). These results demonstrated that the effect of improper temperature on the size distribution of starch panicles is correlated with variety.

### 3.4. Amylose Content

The proportions and properties of amylose are usually tested for estimating the cooking, eating, textural, and nutritional qualities of rice [2]. Therefore, we measured the amylose content (AC) to directly reflect the proportion of amylose. Only 8XR274 (29.01% under NT) could be considered to contain an intermediate amount of amylose, and the other five restorer lines, especially Huazhan (10.42% under NT), were varieties with low amylose content according to the AC classification [37] (Figure 4A). Previous studies have reported that the HT-ripened grains contained decreased levels of amylose [9,20,22]. Our results for R751 and 8XR274 confirmed the negative effect of HT on the amylose content, even though the extent of decline of R751 (decreased by 3.82%) was much lower than that of 8XR274 (decreased by 8.62%) (Figure 4A). The ACs of R313 and YWSM were shown to be unaffected by HT, while HT resulted in a substantial increase in AC in Huazhan (increased by 21.15%) (Figure 4A). Our data identified the ACs of R313, YWSM, and R751 as being less affected by HT. HT caused considerable changes in ACs of 8XR274 and Huazhan.

As reported previously, indoor artificially controlled LT increased the amylose content [8,28]. However, few studies have considered how the naturally LT in the field during grain filling affects the AC of rice grains. In this study, it was shown that AC was reduced by LT for R751 (decreased by 9.55%), R313 (decreased by 8.33%), YWSM (decreased by 7.27%), and 8XR274 (decreased by 3.79%) but increased by LT in Huazhan (increased by 5.77%) (Figure 4A). Our results demonstrated that the naturally LT in the field resulted in a reduction in AC in varieties with intermediate AC but led to an increase in AC in varieties with low AC, such as Huazhan, differently from previous studies [8,28]. Rice grains with low AC tend to have a soft texture and hence a favorable ECQ for East Asians [38]. In this study, it was found that both HT and LT at the grain-filling stage affected AC and thus ECQ.

### 3.5. Alkali Spreading Value and Gel Consistency

In addition to the amylose content, gelatinization temperature (GT) is the main determinant of rice cooking time. Rice grains with a low GT require a relatively shorter time to cook, leading to a softer texture [12]. GT is an amylopectin property. The alkali spreading value (ASV) is widely used as an inverse indicator of the GT of milled rice starch granules [2,33]. To investigate the roles of improper temperature on ASV, we compared the ASV of grain powder of the five restorer lines experiencing LT, NT, or HT. It was shown that neither HT nor LT exerted a significant effect on the ASV of 8XR274 and Huazhan (Figure 4B). LT exerted no significant effect on the ASV of R751, R313, or YWSM (Figure 4B). However, the ASV of R751, R313, and YWSM were decreased by 15.71%, 9.23%, and 14.29%, respectively, upon HT at the filling-stage compared with that under NT (Figure 4B). It was suggested that HT resulted in reduction of the ASV and an increase in GT, likely leading to elongation of cooking time and unacceptable texture for grains of R751, R313, and YWSM. Overall, ASVs of R751, R313, YWSM, and 8XR274 were higher than that of Huazhan under each temperature during grain filling, indicating that these varieties have shorter cooking times and softer rice texture than Huazhan.

Gel consistency (GC) is a routinely used indicator to describe amylopectin properties, reflecting the range of cooked rice textures. GC is ranked as hard (27–40 mm), medium (41–60 mm), or soft (61–100 mm), depending on the horizontal migration of cold rice paste after cooking, cooling, and incubation using test tubes [2]. After grain filling under NT, the GCs of R313 and YWSM grains were hard; and the GCs of R751, 8XR274, and Huazhan grains were medium (Figure 4C). Rice grains with softer GC produced tender cooked grains that remained soft after cooling [2]. The results suggested that cooked rice of R751, 8XR274, and Huazhan would be softer than that of R313 and YWSM. Furthermore, GCs of the grains experiencing HT at the filling stage increased significantly in R751, R313, YWSM, 8XR274, and Huazhan, by 37.50%, 52.94%, 72.50%, 38.74%, and 29.70%, respectively (Figure 4C). A previous study reported that *japonica* rice cultivars had about 3.57–4.74% higher GC when grown under hot conditions (with average field temperature ~30 °C during grain filling in 2013) than under the average field temperature of ~28 °C during grain filling in 2014 [4]. The GC increase from our study was clearly greater, and this may have been due to the differences between *indica* and *japonica* varieties and the experimental designs. Our data confirmed that HT in the field at filling stage resulted in an extensive increase in GC content that may have led to a softer texture of cooked rice. LT at the filling stage did not affect the GC of the tested rice varieties except for the reduction in R313 (decreased by 11.76%) (Figure 4C). The results indicated that LT plays an opposite role in GC than HT, in that it may lead to hard texture of cooked rice in some varieties.

### 3.6. Amylopectin Chain Length Distribution (CLD)

Amylopectin consists of branched starch chains. Starch chain-length distribution (CLD, the distribution of the number of monomer units in chains) is a major determinant of rice grain quality [2]. One commonly held conclusion is that HT easily results in a decrease in short-chain amylopectin but more intermediate- and long-chain amylopectin than under normal conditions, although the range of DP (degree of polymerization) values characterizing the length of the chain was slightly different in previous studies [4,6,24]. Our results based on the naturally HT in the field showed that R751, R313, 8XR274, and Huazhan grains contained amylopectin with reduced amounts of short chains (DP 6–12) and intermediate chains (DP 13–24), whereas these varieties were enriched in long chains, with DP having more than 37 (Table 2). This may have contributed to the higher GCs of these four varieties (Figure 4C) according to a previous study showing that rice grains have softer GC due to a higher proportion of short-chain amylopectin [16]. YWSM was the only variety showing different responses to HT, for its short chains of amylopectin were increased, and its long chains were reduced (Table 2). These results suggested the genetic differences among varieties and the complexity of the mechanism underlying the regulation of HT on amylopectin CLD.

Considering that limited studies have investigated the effect of LT on the distribution of CLD, we examined the CLD of five restorer lines under the naturally LT conditions. The results demonstrated that LT led to increases in short chains (DP 6–12) and intermediate chains (DP 13–24), whereas LT decreased medium–long chains (DP 25–36) and long chains; DP had more than 37 in R751, YWSM, 8XR274, and Huazhan (Table 1). R313 grains showed less sensitivity to LT, including a slight increase in intermediate chain (DP 13–24) only (Table 2). Our results demonstrated for the first time that naturally occurring LT in the field during grain filling played an opposite role to HT in the CLD of amylopectin. The proportion of short amylopectin chains was negatively correlated with the GT, and this affects the cooked rice texture and ECQ [39]. Thus, based on the CLD analyses, the results suggested that HT possibly led to higher GT, a harder texture, and a lower level of stickiness, whereas LT led to lower GT, a softer texture, and a higher level of stickiness in most tested varieties in this study.

### 3.7. Starch Pasting Properties as Determined by RVA Analysis

The viscosity of starch pastes was analyzed through the RVA method, which mimics the cooking process of rice grains for determining physicochemical parameters as indirect indicators of ECQ [16,17]. For example, a low setback (SB) value is associated with softness after cooking, and a high viscosity breakdown (BK) value is related to good palatability [16,17,40].

In our study, the SB for 8XR274 increased due to HT (by 120.89%) and decreased due to LT (by 59.07%) (Figure 5A). The conclusion on the effect of HT on the *indica* rice SB was in contrast to the previous work on *japonica* rice cultivar Koshihikari [23], indicating that the mechanism of HT regulation of the SB in *indica* and *japonica* rice is different and complex. Although the SB for Huazhan was not significantly affected by HT, it decreased under LT conditions (by 92.82%) (Figure 5A). The results revealed the opposite effects of LT and HT on the SB. It was worth noting that the SB values for R751, R313, and YWSM were not affected by either HT or LT (Figure 5A). Thus, R751, R313, and YWSM were restorer lines with relatively stable SB values.

Neither HT nor LT played a role in the regulation of the BK values for R313, YWSM, and Huazhan (Figure 5B). However, the BK for R751 increased under HT conditions (by 80.50%), showing no significant difference under LT conditions (Figure 5B). Both HT and LT reduced the BK for 8XR274 by 55.56% and 46.70%, respectively (Figure 5B). However, as reported previously, the BK for Koshihikari was increased under HT conditions [23]. The effects of HT on the BK were associated with varieties. The results indicated that R313, YWSM, and Huazhan exhibited stable BK values. R751 had stable BK values under LT conditions.

Furthermore, the recovery values (RVs) for R751, R313, and YWSM were unaffected by either HT or LT. However, the RV for 8XR274 was decreased by 60.41% under LT, and the RV for Huazhan was increased by 18.565 under HT (Figure 5C). The peak times (PTs) for R751, R313, and YWSM were not affected by either HT or LT. Interestingly, both HT and LT resulted in 12.04% and 9.98% longer PTs for 8XR274, respectively (Figure 5D). Conversely, both HT and LT led to 11.93% and 10.93% shorter PTs, respectively, for Huazhan (Figure 5D). The different, even opposite effects of improper temperature on the PT value among varieties are unexplained to date. However, there was no doubt that R751, R313, and YWSM had stable RVs and PTs that were not easily affected by temperature. Overall, in terms of the SB, BK, RV, and PT, the restorer lines R751, R313, and YWSM were hardly affected by the ambient temperature.

### 3.8. Water Solubility Index and Swelling Power of Starch Particles

The interaction between starch and water is significant for the processing quality of rice food products. To characterize the influence of improper temperature on the starch–water interaction, we compared the water solubility index and swelling power of the starch particles from the five restorer lines. As shown in Figure 6, HT increased the water solubility of R751 and R313 but decreased that of Huazhan. LT resulted in a reduction in the water solubility of R751, R313, YWSM, and Huazhan (Figure 6). However, the extent of change induced by HT was lesser than that induced by LT. The water solubility of the 8XR274 line did not change upon HT or LT treatment. Regarding swelling power, all varieties exhibited unchanged or slightly changed values under improper temperatures compared with normal conditions (Figure 6). The results suggested that LT decreased the water solubility of most restorer lines except for 8XR274. The improper temperature conditions used in this study only slightly affected the swelling power.

## 4. Conclusions

This study systematically analyzed and compared the effects of naturally occurring HT and LT on five male parent lines of hybrid rice from the following four aspects. Firstly, for the milling quality, LT decreased but HT increased the brown rice rates and milled rice rates of sensitive varieties, whereas both decreased the head rice rates of sensitive varieties. Secondly, for the appearance quality, HT further increased the chalkiness of varieties with high chalkiness accompanied with abnormal starch granule morphology, and LT reduced chalkiness. Thirdly, ACs in most tested varieties were decreased under HT but increased under LT, as reported previously [8,9,20,22,28]. However, the effects of HT and LT on AC were in contrast to the findings of the susceptible variety Huazhan, indicating the complexity of temperature effect mechanism on AC. Fourthly, for the starch pasting properties, in contrast to the reported effect of HT on the *japonica* rice cultivar, HT increased the SB and reduced the BK of 8XR274 [23]. Moreover, the effects of improper temperature on the ASV, GC, CLD of amylopectin, water solubility index, and swelling power of starch particles were shown to be correlated with varieties.

Based on the analysis, we concluded that HT and LT usually played opposite roles in the alteration of grain quality of the susceptible varieties. Notably, R751, R313, and YWSM are superior restorer lines due to their head rice rate, chalkiness degree, chalky rice rate, amylose content, alkali spreading value, and pasting properties. These lines were largely unaffected by either HT or LT. Breeding hybrid rice varieties with these restorer lines is likely to support rice quality stability to mitigate the effects of climate change.

## Figures and Tables

**Figure 1 foods-11-03513-f001:**
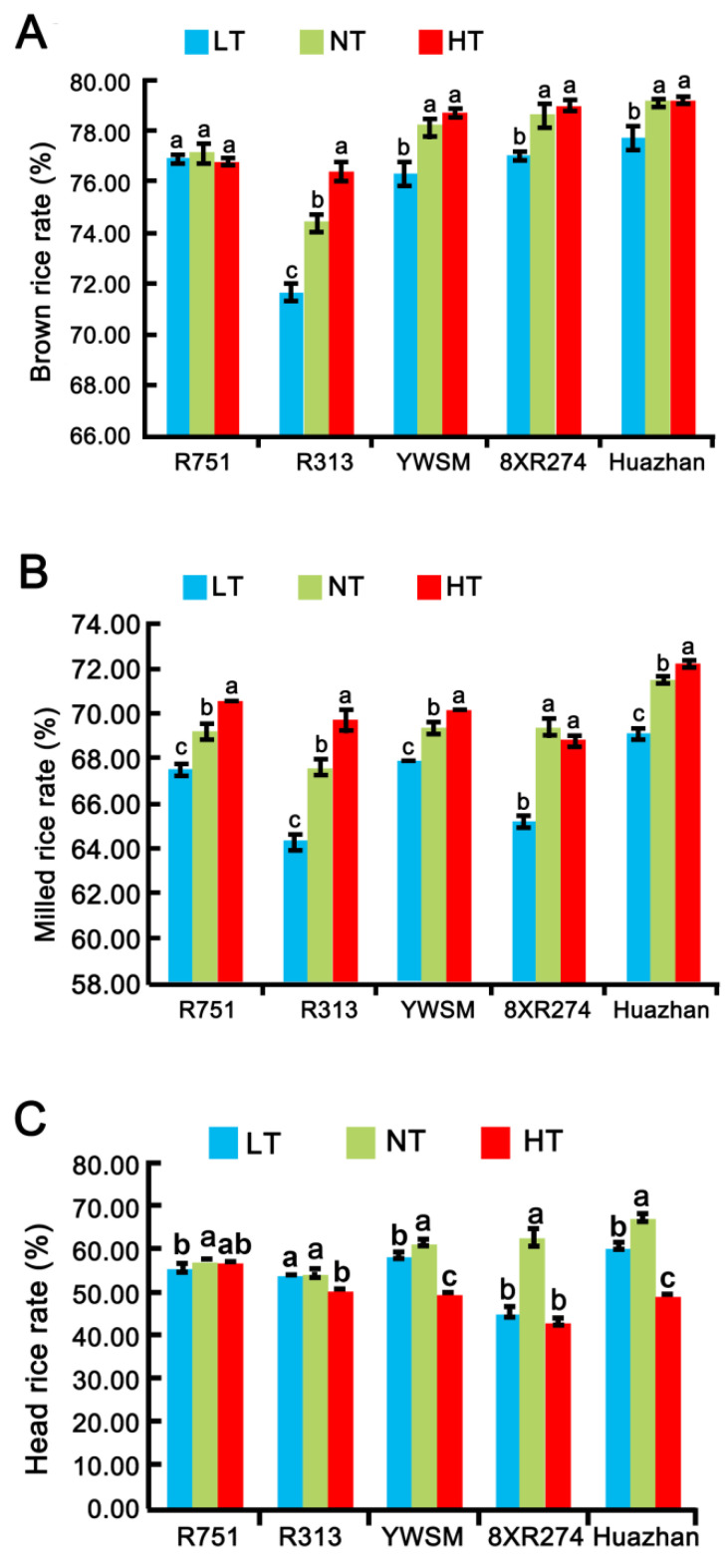
Brown rice rates (**A**), milled rice rates (**B**), and head rice rates (**C**) of different rice varieties affected by improper temperature at the grain-filling stage. LT represents low temperature; HT represents high temperature; NT represents normal temperature control. Data are shown as mean ± standard error of triplicate measurements. Different letters are marked above standard deviations to express significant differences (*p* < 0.05).

**Figure 2 foods-11-03513-f002:**
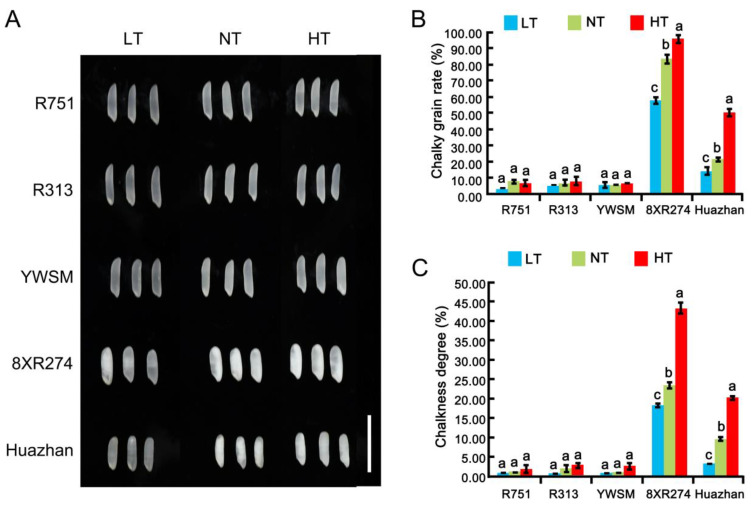
Appearance quality of five restorer lines under different temperatures during the grain-filling stage. (**A**) Photographs of polished grains. Bar = 5 cm. (**B**) Chalky grain rates. (**C**) Chalkiness degree. LT represents low temperature, HT represents high temperature, and NT represents normal temperature controls. Data are shown as mean ± standard error of triplicate measurements. Different letters are marked above standard deviation to express significant differences (*p* < 0.05).

**Figure 3 foods-11-03513-f003:**
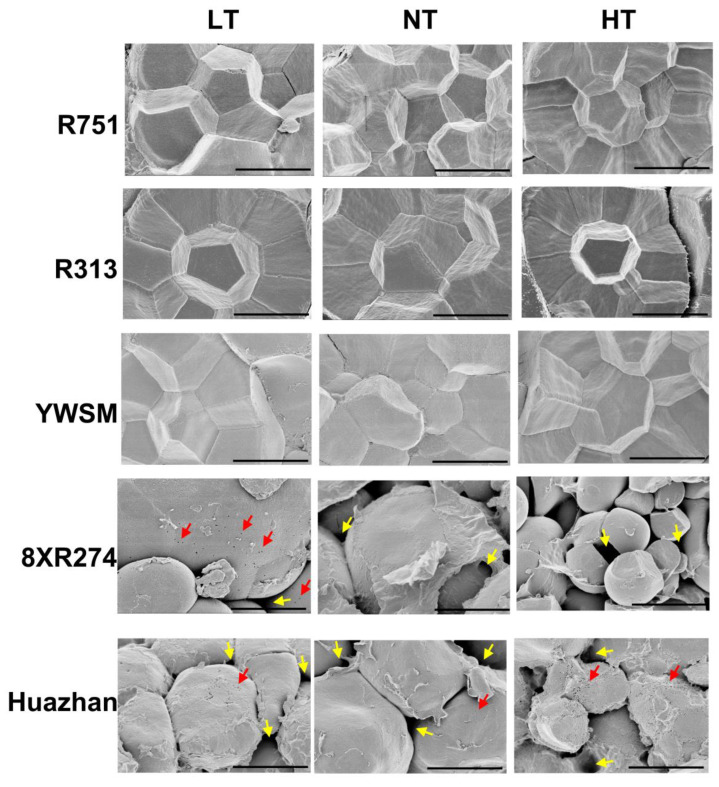
Scanning electron micrographs of starch granules obtained by observing horizontal slices. LT represents low temperature, HT represents high temperature, and NT represents the normal temperature control. Red arrows indicate pits on the surfaces of starch granules. Yellow arrows indicate air spaces in between granules. Bar = 5 μm.

**Figure 4 foods-11-03513-f004:**
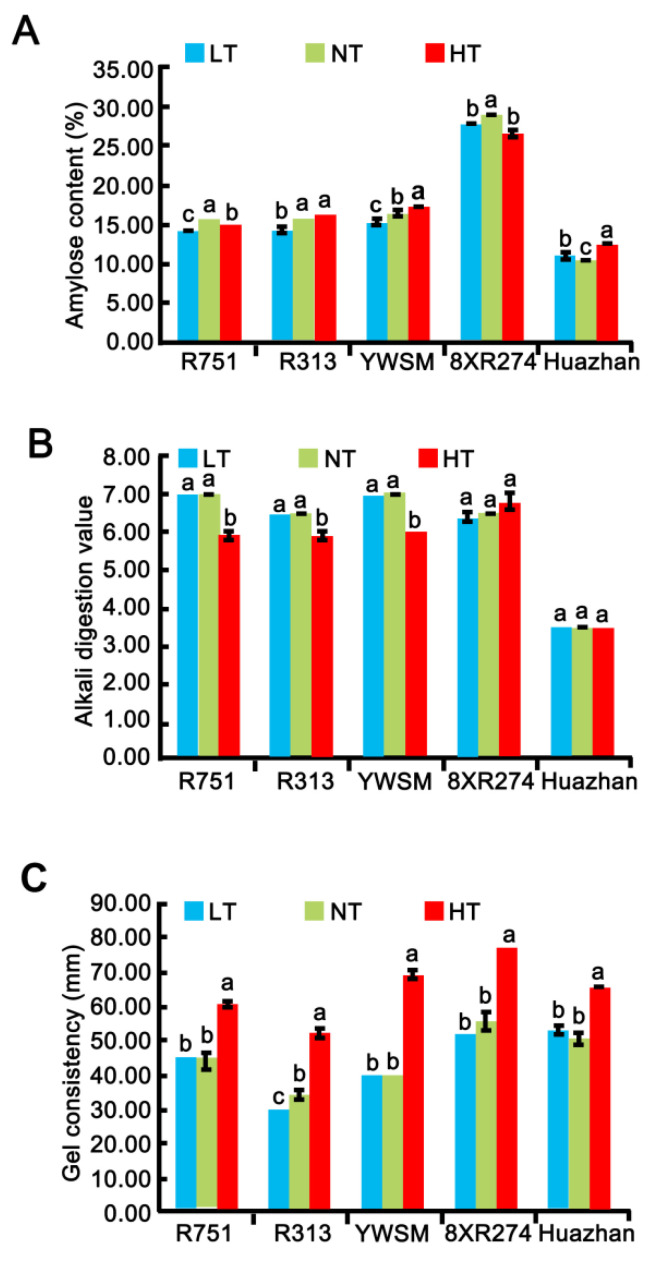
Apparent amylase content (**A**), alkali digestion value (**B**), and gel consistency (**C**) of five restorer lines under LT (low temperature), NT (normal temperature), and HT (high temperature). Data are shown as mean ± standard error of triplicate measurements. Different letters are added above standard deviation to express significant difference (*p* < 0.05).

**Figure 5 foods-11-03513-f005:**
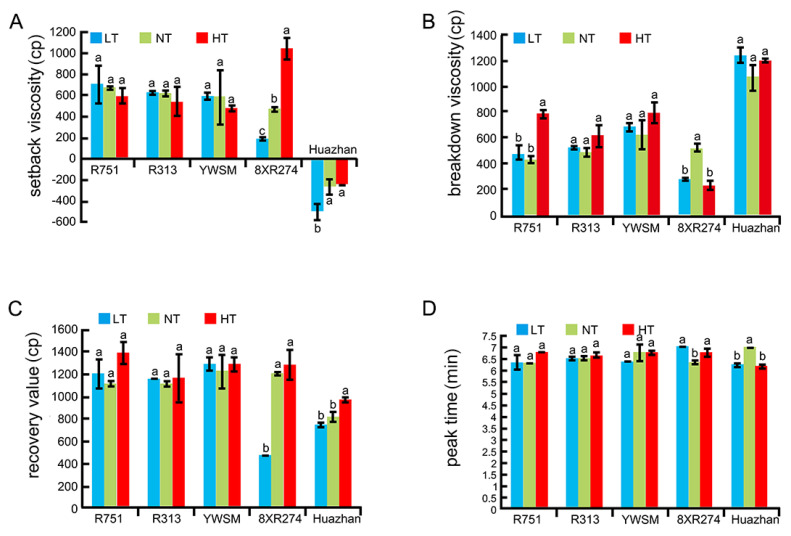
Starch pasting properties as determined by RVA analyses. (**A**) Setback viscosity. (**B**) Breakdown viscosity. (**C**) Recovery value. (**D**) Peak times of the five restorer lines experiencing HT (high temperature), NT (normal temperature), and LT (low temperature) during the grain-filling stage. The unit of starch viscosity is cp, 1 cp = 1/12 RVU (rapid viscosity units). Data are shown as mean ± standard error of triplicate measurements. Different letters are added above standard deviation to express significant differences (*p* < 0.05).

**Figure 6 foods-11-03513-f006:**
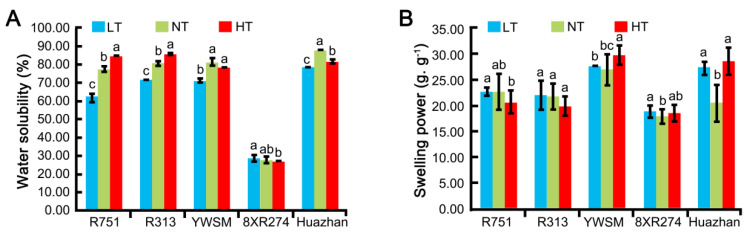
Water solubility index (**A**) and swelling power (**B**) of starch particles of the five restorer lines experiencing HT (high temperature), NT (normal temperature), and LT (low temperature) during the grain-filling stage. Data are shown as mean ± standard error of triplicate measurements. Different letters are added above standard deviation to express significant differences (*p* < 0.05).

**Table 1 foods-11-03513-t001:** Average daily temperature in the field.

Varieties	Average Daily Temperature during Grain-Filling (°C)								
	LT (Low Temperature)			NT (Normal Temperature)			HT (High Temperature)		
	Min	Mean	Max	Min	Mean	Max	Min	Mean	Max
R751	18.3	21.2	26.9	21.6	26.0	33.4	24.8	30.3	39.3
R313	18.3	21.1	26.8	21.7	26.2	33.6	24.8	30.3	39.3
YWSM	18.5	21.5	27.2	22	26.6	34.1	24.8	30.4	39.4
8XR274	18.5	21.5	27.2	21.9	26.4	34.0	25.2	30.3	39.1
Huazhan	18.5	21.5	27.2	21.5	25.8	33.0	24.8	30.4	39.4

**Table 2 foods-11-03513-t002:** Effect of suboptimal temperatures on the chain length distribution of amylopectin.

Restorer Varieties	Temperature in the Grouting Period	Proportions of Amylopectin Chains with Various Degree of Polymerization (DP) (%)			
		6 ≤ DP ≤ 12	13 ≤ DP ≤ 24	25 ≤ DP ≤ 36	DP ≥ 37
R751	LT	28.44 ± 1.50 a	48.97 ± 0.12 a	12.28 ± 0.24 b	10.31 ± 0.93 c
	NT	27.40 ± 1.01 a	47.91 ± 0.22 b	13.11 ± 0.12 a	11.58 ± 0.92 b
	HT	26.91 ± 1.00 a	47.18 ± 0.12 c	13.51 ± 0.33 a	12.40 ± 0.92 a
R313	LT	28.06 ± 1.02 a	48.92 ± 0.10 a	12.43 ± 0.30 a	10.59 ± 3.82 b
	NT	27.89 ± 1.01 ab	48.40 ± 0.13 b	12.75 ± 0.52 a	10.80 ± 4.41 b
	HT	26.84 ± 0.61 b	47.04 ± 0.00 c	13.47 ± 0.54 a	12.65 ± 4.50 a
YWSM	LT	29.03 ± 0.20 a	48.48 ± 0.12 a	12.17 ± 0.12 b	10.32 ± 4.52 c
	NT	27.89 ± 0.12 b	47.47 ± 1.44 a	13.01 ± 0.00 a	11.62 ± 4.50 a
	HT	28.72 ± 0.00 b	47.79 ± 1.61 a	12.82 ± 0.32 a	10.67 ± 4.54 b
8XR274	LT	30.48 ± 1.43 a	48.76 ± 0.62 a	11.75 ± 0.12 b	9.01 ± 4.30 c
	NT	29.01 ± 0.14 ab	47.38 ± 0.60 a	12.68 ± 0.30 a	10.94 ± 4.21 b
	HT	27.98 ± 0.12 b	47.68 ± 0.00 a	12.93 ± 0.00 a	11.40 ± 4.20 a
Huazhan	LT	23.54 ± 0.10 a	53.58 ± 0.31 a	12.12 ± 0.00 c	10.76 ± 3.71 c
	NT	22.78 ± 0.22 ab	53.54 ± 0.12 a	12.47 ± 0.12 b	11.21 ± 3.32 b
	HT	21.63 ± 0.00 b	53.29 ± 0.14 a	13.00 ± 0.00 a	12.07 ± 2.84 a

Data are shown as mean ± standard error of triplicate measurements. Letters represent the differences of three values under three temperature treatments of the same variety in the same column. Different letters follow standard deviations to express significant differences (*p* < 0.05).

## Data Availability

Data are contained within the Appendix A.

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
