# Peer review of "Grain Quality Characterization of Hybrid Rice Restorer Lines with Resilience to Suboptimal Temperatures during Filling Stage"

_foods, 2022, doi:10.3390/foods11213513_

Round 1
Reviewer 1 Report
The manuscript, entitled as “Grain Quality Characterization of Hybrid Rice Restorer Lines with Resilience to Suboptimal Temperature during Filling Stage”, is a typical crop breeding study to explore the effect of temperature on the physicochemical quality of rice. There are numerous lengthy and repetitive narrative descriptions in the article. Inappropriate bolding, underlining, and colloquial words should be improved. The current research topic (rice) is still some way off the edible food (cooked rice).
Apart from the experimental material differences, is there any further innovation in the content of this study from the previous study (cited reference #4)?
4. Characterization of grain quality and starch fine structure of two japonica rice (Oryza Sativa) cultivars with good sensory properties at different temperatures during the filling stage. J. Agric. Food Chem. 2016, 64, 4048-4057.
Sensory experiments on cooked rice require an approved Human Research Permission (IRB). The current content, especially the results and discussions, need to be strengthened. This manuscript is more suitable for publication in cereal and crop related research.
Author Response
The manuscript, entitled as “Grain Quality Characterization of Hybrid Rice Restorer Lines with Resilience to Suboptimal Temperature during Filling Stage”, is a typical crop breeding study to explore the effect of temperature on the physicochemical quality of rice.
- There are numerous lengthy and repetitive narrative descriptions in the article. Inappropriate bolding, underlining, and colloquial words should be improved.
Response: Thanks for your reminder. During the revision of the manuscript, we have used the editing services from LetPub (www.letpub.com) for its linguistic assistance provided from native English speaker. The grammar, wrong symbols and description problems have been modified and marked in red font.
- The current research topic (rice) is still some way off the edible food (cooked rice). This manuscript is more suitable for publication in cereal and crop related research.
Response: Thanks. We can understand your concern about whether our research could match the scope of Foods. Although the research is about rice grains but not cooked rice, the physical and chemical properties of rice grains are closely related with the palatability of rice food. We added a sentence in the last paragraph of the Introduction part on page 2 to emphasize that.
As commented by the academic editor, “This manuscript is about the investigation of "Grain Quality Characterization of Hybrid Rice Restorer Lines with Resilience to Suboptimal Temperature during Filling Stage" and suitable for Foods Journal and our Special Issue of rice.”
Foods has published articles only focusing on rice grains such as “ Tang L.; et al. Food security in China: A brief view of rice production in recent 20 years. Foods, 2022, 11(21), 3224.”
In addition, many researches with results similar to this study that only provide rice quality data but no cooked rice data were published in the journal Food Chemistry with Aims and Scopes similar to Foods. For example: “Yao, D.P.; et al. Influence of high natural field temperature during grain filling stage on the morphological structure and physicochemical properties of rice (Oryza sativa L.) starch. Food Chem. 2020, 310, 125817.” “Zhu, D.; et al. The effects of chilling stress after anthesis on the physicochemical properties of rice (Oryza sativa L) starch. Food Chem. 2017, 237, 936−941.” These two articles were also cited in our manuscript. It is suggested that the current research is suitable for publication in Foods.
- Apart from the experimental material differences, is there any further innovation in the content of this study from the previous study (cited reference #4)?
Reference #4. Characterization of grain quality and starch fine structure of two japonica rice (Oryza Sativa) cultivars with good sensory properties at different temperatures during the filling stage. J. Agric. Food Chem. 2016, 64, 4048-4057.
Response: Thank you for the nice question. Compared with the research cited as reference 4, our study has the following innovations.
(1). The previous study only examined the sensitivity of grain quality of 2 conventional rice varieties to high temperature. Our study examined the sensitivity of 5 restorer lines to both high temperature and low temperature stresses. We studied the effects of not only high temperature but also low temperature. This point was emphasized at the end of 4th paragraph in Introduction on page 2 of the revised manuscript.
(2). In the previous study, the temperature treatment was carried out in different years, with 2013 as the high temperature treatment and 2014 as the normal temperature treatment. In comparison, we sowed a batch of materials every 14 days in the same field from April 11th to June 20th in 2020. Three batches of materials that have experienced low temperature, normal temperature and high temperature at the grain-filling stage are then selected from the six batches for analysis. It is possibly that the influence of uncontrollable factors such as field water and fertilizer management, diseases and pests, etc in the natural environment in the same year is smaller than that of the experiments in two independent years.
(3). Milling quality is closely related with the edible food yield. In our study, low temperature during grain filling stage led to the decline while high temperature resulted in the increase of milled rice rate; however, low temperature had a greater impact for most tested varieties. Our study also show that rice varieties R751, R313 and YWSM harbor good milling quality upon high or low temperature treatments at grain-filling stage. However, milling quality was not investigated in the previous study cited reference #4.
(4). Both our work and the previous published work (reference #4) have examined the morphological characteristics of starch granules under high temperature stress through electronic scanning microscope. However, we directly scanned the horizontal slices of grains to observe the morphological characteristics of the starch granules, whereas the previous work was to detect the starch particle characteristics after the rice grains were ground into powder. Our results directly reflect the effect of temperature on the morphology of starch granules, especially the relatively accurate characteristics of the gaps between starch granules in rice grains are accurate. In the revised version, we clearly emphasized this point in Results and Discussion 3.3 on page 9.
- Sensory experiments on cooked rice require an approved Human Research Permission (IRB).
Response: Thanks for the reminder. But we did not conduct sensory experiments on cooked rice in this study and thus there is no need for an approved Human Research Permission.
- The current content, especially the results and discussions, need to be strengthened.
Response: Thanks for the advice! We have made a comprehensive revision of the whole “Results and Discussion” part (marked in red font), hoping that the results will be clearer and the discussion more organized.

Reviewer 2 Report
The article is dedicated to the qualitative characterization of five rice lines, placed at naturally unfavorable temperatures during the filling period in field conditions.
The study related some physical and chemical characteristics. The work is clear and well written, the results are explained clearly and effectively. I only require minor changes:
- Move the temperature table, found in the additional files, in the main text between materials and methods.
- Add in the introduction what can be the practical implications of the information obtained with updated bibliographic references.
Author Response
The article is dedicated to the qualitative characterization of five rice lines, placed at naturally unfavorable temperatures during the filling period in field conditions. The study related some physical and chemical characteristics. The work is clear and well written, the results are explained clearly and effectively. I only require minor changes:
- Move the temperature table, found in the additional files, in the main text between materials and methods.
Response: Thanks for the valuable advice. In the revised manuscript, the table listing the temperature treatments was moved in the main text and designated as Table 1.
- Add in the introduction what can be the practical implications of the information obtained with updated bibliographic references.
Response: Thanks for the suggestion. On page 2 of the revised manuscript, “These studies suggest that the adjustment of sowing date can reduce the negative influence of temperature stress on rice quality” was added to state the potential practical implications of recent studies. “Changes in physical and chemical properties of rice grains caused by temperature stress are likely to lead to deterioration of the taste of various rice foods such as cooked rice, rice noodles, steamed rice sponge cake, and rice crust. The aim of this work was to identify the restorer lines that have excellent grain quality and that are insensitive to both high temperature and low temperature. The results will contribute to create germplasm resources where the taste of rice is unaffected by temperature stress” was added at the end of the Introduction Part to state the potential practical implications of our study.

Reviewer 3 Report
Review on manuscript: foods-1963469
Grain Quality Characterization of Hybrid Rice Restorer Lines with Resilience to Suboptimal Temperature during Filling Stage
by Xuedan Lu, Lu Wang, Yunhua Xiao, Feng Wang, Guilian Zhang, Wenbang Tang and Huabing Deng
submitted to Foods
In the manuscript submitted for comments, the authors studied the effect of temperature on the responses of five restorer rice lines at grain-filling stage by conducting field experiments.
Generally the manuscript is prepared correctly, however I am not convinced that the authors chose the journal correctly. The link between the research objective and the obtained results with food is not properly emphasized.
Abstract – it should refer to specific results to a greater extent,
Keywords – high temperature and low temperature do not reflect the content of the manuscript,
page 2 – the literature review should end with a clearly formulated research objective and not a presentation of the methods used, maybe the title and purpose of the research should refer to the starch which the authors have characterized?
page 4 – the names of producers and countries of origin of the devices used should be given,
paragraph 2.7 – the description of the methods should be consistent with the title of the paragraph, the description of the methods should be extended,
paragraph 2.10 – what were the factors in the analysis of variance?
paragraph 3.6 – the methodology lacks information on the CLD analysis,
Table 1 legend – Different letters … in rows or in columns?
Figure 5 – lack of units,
Figure S2 – should be moved to main text,
References – should be formatted as required for authors.
Author Response
In the manuscript submitted for comments, the authors studied the effect of temperature on the responses of five restorer rice lines at grain-filling stage by conducting field experiments.
- Generally the manuscript is prepared correctly, however I am not convinced that the authors chose the journal correctly. The link between the research objective and the obtained results with food is not properly emphasized.
Response: Thanks for your comments. We can understand your concern about whether our research could match the scope of Foods. Although the research is about rice grains but not cooked rice, the physical and chemical properties of rice grains are closely related with the palatability of rice food and also food security.
As commented by the academic editor, “This manuscript is about the investigation of "Grain Quality Characterization of Hybrid Rice Restorer Lines with Resilience to Suboptimal Temperature during Filling Stage" and suitable for Foods Journal and our Special Issue of rice.”
Foods has published articles only focusing on rice grains such as “Tang L.; et al. Food security in China: A brief view of rice production in recent 20 years. Foods, 2022, 11(21), 3224”. In addition, many researches with results similar to this study that only provide rice quality data but no cooked rice data were published in the journal Food Chemistry with Aims and Scopes similar to Foods. For example: “Yao, D.P.; et al. Influence of high natural field temperature during grain filling stage on the morphological structure and physicochemical properties of rice (Oryza sativa L.) starch. Food Chem. 2020, 310, 125817”. “Zhu, D.; et al. The effects of chilling stress after anthesis on the physicochemical properties of rice (Oryza sativa L) starch. Food Chem. 2017, 237, 936−941.” These two articles were also cited in our manuscript. It is suggested that the current research is suitable for publication in Foods.
To properly emphasize the link between the research objective and the obtained results with food, we added a sentence “Changes in physical and chemical properties of rice grains caused by temperature stress are likely to lead to deterioration of the taste of various rice foods such as cooked rice, rice noodles, steamed rice sponge cake, and rice crust” at the last paragraph of the Introduction. Moreover, in the Result and Discussion part, we have explained the relationship between the measured indicators and the quality of rice food. For instance, amylose content with eat and cooking quality in 3.4 section, Alkali Spreading Value and Gel Consistency with cooking time and texture of cooked rice in 3.5 section, starch viscosity with good palatability in 3.7 section, the water solubility and swelling power with the processing quality of rice food products in 3.8 section.
- Abstract – it should refer to specific results to a greater extent.
Response: Thanks for your valuable advice! We have changed the last sentences of the Abstract as “Breeding hybrid rice with adverse-temperature-tolerant restorer lines can not only ensure high yield via heterosis but also produce superior grain quality. This can ensure the quantity and taste of rice as a staple food in the future when extreme temperatures occur increasingly frequently.”
- Keywords – high temperature and low temperature do not reflect the content of the manuscript.
Response: Thanks for your valuable advice! The two keywords have been replaced as “temperature stress insensitivity”.
- page 2 – the literature review should end with a clearly formulated research objective and not a presentation of the methods used, maybe the title and purpose of the research should refer to the starch which the authors have characterized?
Response: Thanks for your advice! We have already revised the last part of the literature review with “Changes in physical and chemical properties of rice grains caused by temperature stress are likely to lead to deterioration of the taste of various rice foods such as cooked rice, rice noodles, steamed rice sponge cake, and rice crust. The aim of this work was to identify the restorer lines that have excellent grain quality and that are insensitive to both high temperature and low temperature. The results will contribute to create germplasm resources where the taste of rice is unaffected by temperature stress.
Our research is about not only the physical and chemical properties of starch, but also the milling quality and appearance quality. So we still think it is more appropriate to retain the original title rather than change it to starch.
- Page 4 – the names of producers and countries of origin of the devices used should be given,
Response: Thanks for your kind reminder! All the manufacturer's name and location for all specialized equipment have been provided in the Methods.
- paragraph 2.7 – the description of the methods should be consistent with the title of the paragraph, the description of the methods should be extended.
Response: Thanks for your kind reminder! The description of the measurement methods of AC, GC and ASV has been extended on page 7 as marked in red font.
- paragraph 2.10 – what were the factors in the analysis of variance?
Response: Thanks for the question. We have done the Two-way analysis of variance, with three fixed factors including rice varieties, temperature treatments and the interaction between rice varieties and temperature treatments. Because the interaction is significant, the result for each variety was shown.
- paragraph 3.6 – the methodology lacks information on the CLD analysis,
Response: Thanks for your reminder! The methods for CLD analysis was added in 2.6 on page 4.
- Table 1 legend – Different letters … in rows or in columns?
Response: Thanks for your question! Different letters represent the difference of three values under three temperature treatments of the same variety in the same column.
- Figure 5 – lack of units
Response: Thanks for your question! The unit of starch viscosity is cp, 1 cp = 1/12 RVU (rapid viscosity units). The unit of peak time is minute. And Figure 5 and its legends have been revised.
- Figure S2 – should be moved to main text
Response: Thank you! We have changed the position of Figure S2 as your advice.
- References – should be formatted as required for authors.
Response: Thank you for the reminder. We have carefully checked the format of all references and revised them based on the latest articles published in the journal.

Reviewer 4 Report
This study systematically analyzed and compared the effect of naturally HT and LT on five male parent lines of hybrid rice from the following three aspects.
Firstly, for the milling quality, LT decreased but HT increased brown rice rate and milled rice rate of sensitive varieties, whereas both of them decreased head rice rate of sensitive varieties.
Secondly, for the appearance quality, HT further increased chalkiness of varieties with high chalkiness accompanied with abnormal starch granule morphology while LT reduced chalkiness.
Thirdly, for AC content, HT resulted in a substantial increase of AC in the susceptible variety Huazhan while LT decreased AC. It showed totally different effect of HT and LT on AC in previous studies.
Fourthly, for the starch pasting properties, opposite to the reported effect of HT on japonica rice cultivar, HT increased the SB and reduced the BK of 8XR274 . Moreover, the effect of improper temperature on ASV, GC, CLD of amylopectin, water solubility index and swelling power of starch particles were shown to be correlated with varieties.
Based on the analysis, we draw a conclusion that HT and LT usually played opposite roles on the alternation of grain quality of the susceptible varieties. But more notably, R751, R313 and YWSM are superior restorer lines because their head rice rate, chalkiness degree, chalky rice rate, amylose content, alkali spreading value, and pasting properties exhibited little alternation to either HT or LT.
Breeding hybrid rice varieties with these restorer lines is likely to support rice quality stability to mitigate the effects of climate change.

Author Response
Response: Really thank you very much for your positive comments on this work

Round 2
Reviewer 1 Report
Thanks to the authors for their responses to my question, some minor flaws (and more in the draft) are listed below,
What is NaN3 in "Sodium acetate (50 μL, 0.6M, pH 4.4), NaN3 (10 μL, 2% w/v) ...."?
Avoid vertical dividing lines in Table 1.
Too much lengthy descriptions of experimental procedures are inappropriate in paragraphs 2.6. and 2.7.
Even the consumption of ordinary daily cereals should be approved by the applicant's body research review.
I have no further questions in this manuscript.
Author Response
- What is NaN3 in "Sodium acetate (50 μL, 0.6M, pH 4.4), NaN3 (10 μL, 2% w/v) ...."?
Response: Thank you for the question and that also reminds me to change all the chemical formula of chemicals in the manuscript into English name. NaN3 is actually the chemical formula of sodium azide.
- Avoid vertical dividing lines in Table 1.
Response: Thanks for your suggestion! The vertical dividing lines were erased in Table 1.
- Too much lengthy descriptions of experimental procedures are inappropriate in paragraphs 2.6. and 2.7.
Response: Thanks! According to your suggestion, we have simplified the description of the methods for CLD, AC and GC measurements (marked in green font in 2.6 and 2.7 parts). However, another reviewer have suggested us to extend the description of methods in 2.6 and 2.7 in the 1st round of review, the current description might be appropriate.
- Even the consumption of ordinary daily cereals should be approved by the applicant's body research review.
Response: Thanks for the comments. We agree that the consumption of ordinary daily cereals need to be approved by the applicant’s body research review. But in the current study, we did not involve the consumption of rice by people. We investigated the appearance, milling quality and physicochemical properties of rice starch. During the whole experimental process, rice was not processed into food or eaten by anyone. Based on this, we believe that the current research does not need to require Human Research Permission. However, we still appreciate your reminder. If food tasting is involved in the future research, we will definitely apply for permission before the experiment.
